# Polysaccharides from *Hericium erinaceus* Fruiting Bodies: Structural Characterization, Immunomodulatory Activity and Mechanism

**DOI:** 10.3390/nu14183721

**Published:** 2022-09-09

**Authors:** Yang Yang, Jihong Li, Qing Hong, Xuehong Zhang, Zhenmin Liu, Tiehua Zhang

**Affiliations:** 1State Key Laboratory of Dairy Biotechnology, Shanghai Engineering Research Center of Dairy Biotechnology, Dairy Research Institute, Bright Dairy & Food Co., Ltd., Shanghai 200436, China; 2Department of Food Quality and Safety, College of Food Science and Engineering, Jilin University, 5333 Xi’an Road, Changchun 130062, China; 3State Key Laboratory of Microbial Metabolism, School of Life Sciences and Biotechnology, Shanghai Jiao Tong University, 800 Dongchuan Road, Shanghai 200240, China

**Keywords:** *Hericium erinaceus* polysaccharides, NMR spectroscopy, structural analysis, immunomodulation activity

## Abstract

Five fractions from crude *Hericium erinaceus* polysaccharides (HEPs), including HEP-1, HEP-2, HEP-3, HEP-4 and HEP-5, were obtained through column chromatography with a DEAE Cellulose-52 column and Sephadex G-100 column. The contents of total carbohydrates and uronic acid in HEPs were 53.36% and 32.56%, respectively. HEPs were mainly composed of Fuc, Gal and Glu in a molar ratio of 7.9:68.4:23.7. Its chemical structure was characterized by sugar and methylation analysis, along with ^1^H and ^13^C NMR spectroscopy. HEP-1 contains the backbone composed of (1→6)-linked-galactose with branches attached to O-2 of some glucose. The immunological activity assay indicated that HEP-1 significantly promoted the production of nitric oxide, interleukin-6, interleukin-10, interferon-γ and tumor necrosis factor-α and the phosphorylation of signaling molecules. Collectively, these results suggested that HEP-1 could improve immunity via NF-κB, MAPK and PI3K/Akt pathways. *Hericium erinaceus* polysaccharides might be explored as an immunomodulatory agent for use in dietary supplements.

## 1. Introduction

*Hericium erinaceus* is a type of well-known edible and medicinal fungus belonging to the family *Hericiaceae* in the order *Russulales*. It has many functional benefits such as antitumor activities [1], immunomodulatory effects [2], antioxidant properties [3], cytotoxic effects [4] and neuron growth promoting effects [5]. Two polysaccharide fractions, termed HPA and HPB, were isolated from the fruiting body of *Hericium erinaceus Pers*. The polysaccharide of HPA consists of Glc, Gal and Fuc in the ratio 1:2.1:0.4, and HPB consists of Gal and Glc in the ratio of 1:11.5. HPA contains the backbone composed of (1→6)-linked-galactose with branches attached to O-2 of galactose. HPB contains the backbone composed of (1→6)-linked-glucose with branches attached to O-3 of glucose [6]. Six polysaccharides (FI0-a, FI0-a-α, FI0-a-β, FI0-b, FII-1, FIII-2b) isolated from the fresh fruiting bodies of *H. erinaceus* were composed of xylans, glucoxylans, heteroxyloglucans and galactoxyloglucans. FI0-a-α and FI0-a-β had β-(l→3) and (1→6) glucan chains. FII-1 and FIII-2b had β-(1→3) and (l→6) glucosidic bonds [7]. Three heteropolysaccharides termed HEPA1, HEPA4, HEPB2 were isolated from the mycelium of *H. erinaceus*. They mainly consisted of glucose. Moreover, HEPA4 is a heteropolysaccharide peptide with a 3.8% of protein fraction, and an α-linkage was the main glucosidic linkage configuration among all the homogeneous monosaccharides [8]. *Hericium erinaceus* polysaccharides have differences in structures due to different origin and growing environments. Therefore, *Hericium erinaceus* polysaccharides grown in Jilin Province were selected and investigated.

In recent years, there have been many reports on the immunomodulatory effect of polysaccharides. Polysaccharides from Lotus (*Nelumbo nucifera Gaertn*) root residues enhanced immune function both in vitro and in vivo via inducing nuclear translocation of activator protein-1 and NF-κB [9]. A new polysaccharide containing N-acetylglucosamine was isolated from *Morchella importuna* fruiting bodies. It exerted immune-potentiating effects by stimulating macrophage function, promoting phagocytosis and increasing secretion of NO, TNF-α and IL-6 through the TLR4/JNK and Akt/NF-κB signaling pathways in RAW264.7 cells [10]. In addition, mushroom polysaccharide can promote immune response in murine macrophage cells via the TLR/NF-κB pathway [11]. The polysaccharides extracted from the fresh fruiting bodies of *Hericium erinaceus* induced the proliferation of lymphocytes and enhanced the expression of inflammatory cytokines IL-6, TNF-α and IL-1β secreted by THP-1 macrophages [12]. Therefore, it is necessary to further estimate the signaling pathways of immunomodulatory effect of *Hericium erinaceus* polysaccharides. In addition, immune cells were treated by polysaccharides directly in these reports above, but in fact, when polysaccharides enter the intestine, they firstly meet the intestinal mucosa, i.e., the intestinal epithelial cell, and then may meet the immune cells, e.g., the macrophages. Therefore, we built a Caco-2/RAW264.7 co-culture system to simulate the process.

In the current study, we intended to resolve the structure of polysaccharide fractions from *Hericium erinaceus* by high performance liquid chromatography, gas phase mass spectrometer, Fourier transform infrared spectroscopy and nuclear magnetic resonance. We further compared the immunomodulatory activities of different polysaccharide fractions and investigated the immunomodulatory mechanisms using cellular experiments.

## 2. Materials and Methods

### 2.1. Materials and Reagents

The fruiting body of *Hericium erinaceus* was purchased from Jilin Jiaohe Songshan Food Co., Ltd. (Jiaohe, China). Dextran standards (670, 410, 270, 80, 25, 12 and 5 kDa) were purchased from Sigma Chemical Co. (St. Louis, MO, USA). Monosaccharide standards, including arabinose, mannose, fucose, xylose, rhamnose, galactose and glucose were purchased from Sigma Chemical Co. (St. Louis, MO, USA). Cell Counting Kit-8 was obtained from APExBIO Technology LLC (Houston, TX, USA). Neutral red staining solution, nitric oxide assay kit and BCA kit were obtained from Beyotime Biological Technology Co., Ltd. (Shanghai, China). Western Blot assay kit, SDS-PAGE kit, ELISA kits of TNF-α, IL-6, IL-10 and IFN were obtained from Boster Biological Technology Co., Ltd. (Wuhan, China). Anti-ERK, anti-phospho ERK, anti-JNK, anti-phospho JNK, anti-p38 MAPK, anti-phospho p38 MAPK, anti-Akt, anti-phospho Akt, anti-IkBα, anti-p65, anti-β-actin and anti-Histone H2A were purchased from ABclonal Biotechnology Co., Ltd. (Wuhan, China). Inhibitors of BAY11-7082, SB203580, SP600125, PD98059 and LY294002 were purchased from Selleck Chemicals (Shanghai, China).

### 2.2. Isolation and Purification of Polysaccharides

The *Hericium erinaceus* polysaccharides were extracted by hot-water extraction technique [13]. Dried Hericium erinaceus was ground into power. Distilled water (at a ratio of 1:10 *w*/*v*) was added to the power and was boiled for 8 h and was then cooled to room temperature. The mixture was filtered in vacuum, and concentrated in a rotary evaporator under a low pressure at 45 °C. The concentrated solution was added with a four-fold volume of ethanol to the concentrate to precipitate the polysaccharides, overnight. The precipitate was collected by centrifugation and washed successively with deionized water.

The solution of the crude polysaccharides was treated with 5% trichloroacetic acid for 30 min to remove remaining proteins and centrifuged at 10,000 rpm for 30 min. The supernatant was concentrated in a rotary evaporator in vacuum and then was dialyzed in a 500–1000D dialysis bag. The solution in the dialysis bag was collected and concentrated in a rotary evaporator in vacuum and then lyophilized, affording HEPs.

The crude polysaccharides (100 mg) were dissolved in deionized water (20 mL) and filtered by a 0.45 μm pore size membrane. The polysaccharide solution was loaded on a DEAE Cellulose-52 column (2.6 × 30 cm) and eluted with deionized water and gradient NaCl solutions (0.05, 0.1, 0.2, 0.3 and 0.5 M NaCl) at a flow rate of 1.00 mL/min. All the eluates were collected by tubes (10 mL per tube). Total carbohydrate content of each tube was measured at 490 nm by phenol-sulfuric acid [14], drawing an elution curve. Samples were collected at the best absorption-peak sections and then were concentrated under a low pressure and lyophilized to obtain the DEAE Cellulose-52 separation components of polysaccharide.

The polysaccharide sample (5 mg/mL) was loaded on a Sephadex G-100 column (2.6 × 60 cm) and eluted with deionized water at a flow rate of 0.25 mL/min. All the eluates were collected by tubes (10 mL per tube). Total carbohydrate content of each tube was measured at 490 nm by the phenol–sulfuric acid [14], drawing an elution curve. The best absorption peak in sections was selected to collect samples. Through decompression concentration and freeze-drying, the Sephadex G-100 separation components of polysaccharide were obtained (named HEP-1, HEP-2, HEP-3, HEP-4, HEP-5).

### 2.3. General Analytical Methods

#### 2.3.1. Determination of Chemical Properties

The total carbohydrate content of HEPs was determined by the phenol–sulfuric acid method [14]. The uronic acid content was measured by the carbazole−sulfuric acid method [15]. The sulfate content was determined according to the referenced method [16]. Ultraviolet visible spectrum was used for evaluating whether samples contain impurities such as nucleic acid or protein [17].

#### 2.3.2. Determination of Homogeneity and Molecular Weight

Homogeneity and the number-average molecular weight of HEPs were determined on a UPLC system equipped with a Waters Ultrahydrogel linear column (7.8 mm × 300 mm) and a refractive index (RI) detector. The sample solution (20 µL) was filtered by a 0.45 µm membrane, with water as the mobile phase at a flow rate of 0.3 mL/min. The linear regression was calibrated with Dextran standards (670, 410, 270, 80, 25, 12 and 5 kDa) [18].

#### 2.3.3. Monosaccharides Composition

The samples were degraded into monosaccharides by hydrolysis with 2 M trifluoroacetic acid (TFA) at 121 °C in a sealed-tube for 6 h. After aldononitrile acetate derivatization, the acetylated derivatives were analyzed by the Agilent 6890 GC-MS system (Agilent Technologies Inc., Santa Clara, CA, USA) equipped with a Nitrogen Phosphorus Detector (NPD) and an Agilent 19091U-433 column (30 m × 0.25 mm × 0.25 µm). Column temperature was programmed from 80 °C/min, for 3 min, and then increased to 280 °C, at a rate of 20 °C/min. Injection temperature: 250 °C; detector temperature: 250 °C; nitrogen was used as the carrier gas and set at a constant flow rate of 20.0 mL/min with a split ratio of 1:30 [19].

#### 2.3.4. Fourier Transform Infrared (FT-IR) Spectroscopy

The samples were ground with KBr powder and then pressed into tablets. FT-IR spectroscopy was recorded by using a Shimadzu Fourier transform (FT)/IR IRPRESTIGE-21 spectrophotometer in the frequency range of 4000–500 cm^−1^ with the resolution of 4.0 cm^−1^ and 320 scans co-addition [20].

#### 2.3.5. Methylation Analysis

Methylation analysis was performed according to a previous study [21]. Five mg of polysaccharides were treated with 3 mL of 90% aqueous formic acid for 6 h at 100 °C in a sealed tube. After removing the formic acid, the per-O-methylated product was hydrolyzed with 2 M TFA (2 mL) at 100 °C for 3 h and concentrated to dryness. The residue was reduced with sodium borohydride, and neutralized with acetic acid, then acetylated to give a mixture of partially O-methylated alditol acetates. The resultant elutants were qualitatively and quantitatively analyzed by GC–MS on a Shimadzu GCMS-QP 2010 instrument (Kyoto, Japan) with RXI-5 SIL MS fused-silica capillary column.

#### 2.3.6. Nuclear Magnetic Resonance (NMR) Spectroscopy

The sample (30 mg) was dissolved in 0.6 mL of D_2_O. ^1^H and ^13^C NMR spectra were recorded at 60 °C by a Bruker AV-600 spectrometer (Bruker, Rheinstetten, Germany). Chemical shifts are expressed in ppm using acetone as internal standard [22].

### 2.4. Immunomodulatory Activity Analysis

#### 2.4.1. Cell lines and Culture

Murine macrophage cell line (RAW264.7) and human intestinal epithelial cell line (Caco-2) (Shanghai Institutes for Biological Sciences, Chinese Academy of Sciences) were cultured in an DMEM medium supplemented with 10% (*v*/*v*) fetal bovine serum and penicillin–streptomycin solution at 37 °C in a humidified incubator with 5% CO_2_. Cells were sub-cultured in the same medium under the same conditions for further experiments.

#### 2.4.2. Cytotoxicity Assay by CCK-8

The RAW264.7 cells were seeded at a concentration of 1 × 10^5^ cells/mL in a volume of 100 μL in 6-well plates. Cells were incubated with HEP-1, HEP-2, HEP-3, HEP-4, HEP-5 (0, 25, 50, 100, 200, 400 and 800 μg/mL). After 24 h, each well was added with 10 μL of CKK-8 solution and incubated for another 2 h. Absorbance was detected by microplate ELISA reader at 450 nm.

The cell viability was calculated as follows:$$\varphi =\frac{\mathrm{As}-\mathrm{Ab}}{\mathrm{Ac}-\mathrm{Ab}}\times 100\%$$
where As is the absorbance value of sample group, Ab is the absorbance value of blank group and Ac is that of control group [23].

#### 2.4.3. Nitric Oxide Production

RAW264.7 cells (1 × 10^5^ cells/well) were incubated with HEP-1, HEP-2, HEP-3, HEP-4, HEP-5 (0, 25, 50, 100, 200, 400 and 800 μg/mL). After 24 h, 50 μL of the supernatants were isolated and mixed with an equivalent volume of Griess reagent. 0.1% *N*-(1-naphthyl) ethylenediamine dihydrochloride solution and 1% sulfanilamide in 5% concentrated H_3_PO_4_ solution were mixed to afford Griess reagent. After 10 min, the absorbance was measured by microplate ELISA reader at 540 nm. Nitrite concentration was calculated from an NaNO_2_ standard curve and was expressed as nmol per 1 × 10^5^ cells.

#### 2.4.4. Assay for Phagocytic Activity

RAW264.7 cells (1 × 10^5^ cells/well) were incubated in the culture medium in the absence (control) or presence of HEP-1, HEP-2, HEP-3, HEP-4, HEP-5 (0, 25, 50, 100, 200, 400 and 800 μg/mL). After 24 h, the cells were added with neutral red dye and incubated for 1 h and the neutral red dye was removed. The cells were washed three times with the warm PBS and was added with acetate–ethanol cytolytic solution. After 2 h at 4 °C, the absorbance was detected by a microplate ELISA reader at 540 nm.

#### 2.4.5. Caco-2/RAW264.7 Co-Culture System and Cytokine Secretions in RAW264.7 Cells

After cell thawing, Caco-2 cells culture medium was changed every other day until the cell confluence reached 90% (about 4–6 days) and then sub-cultured. Caco-2 cells were seeded at 1 × 10^5^ cells per well onto the upper compartment of Transwell insert plates at 35 °C in a 5% CO_2_ incubator. The morphology of cells was observed under light microscope on day 1, day 5 and day 15. On the 15th day, when Caco-2 cells were uniformly distributed, well-defined and arranged in a dense monolayer under the light microscope, RAW264.7 cells were seeded at 1 × 10^5^ cells per well onto the lower compartment of Transwell insert plates. After 24 h of co-culture, Transwell insert plates were added into the upper compartment by various concentrations of HEP-1 (0, 50, 100, 200 and 400 μg/mL) and lipopolysaccharides (LPS) (1 μg/mL) (positive control). After co-culturing for additional 24 h, the cytokines in the lower compartment were assayed according to the cytokine ELISA protocol from the manufacturer’s instructions [24].

#### 2.4.6. Western Blotting

RAW264.7 cells were seeded in 6-well culture plates at a concentration of 1 × 10^6^ cells/mL for 24 h and were then treated with or without specific inhibitors (NF-κB pathway inhibitor 10 μM BAY 11-7082, MAPK pathway inhibitor 20 μM SB203580, 10 μM SP600125, 20 μM PD98059 or Akt pathway inhibitor 20 μM LY294002) for 1 h. After 1 h, cells were treated for 3 h with various concentrations of HEPs. Next, the cells were washed three times with ice-cold PBS and centrifuged at 5000 rpm for 3 min, and the supernatant was removed. Cells were lysed with RIPA for 1 min on ice, and the cell lysates were centrifuged at 4 °C for 10 min (10,000 rpm). The protein levels in the supernatant were determined using the BCA assay. The proteins (40 μg) were separated by electrophoresis of a 10% sodium dodecyl sulfate-polyacrylamide gel (SDS-PAGE) and transferred onto a PVDF membrane. The membrane was blocked with 5% skim milk in TBST for 1.5 h and then detected by the ECL Western blot detection kit to expose signals to X-ray films. Protein expression was analyzed using the software Quantity One (version 4.6, Bio-Rad, Hercules, CA, USA) [25].

#### 2.4.7. Statistical Analysis

Data are presented as the mean ± standard deviation (SD). The significance of difference was evaluated with one-way ANOVA, followed by Student’s *t*-test to statistically identify differences between the control and treated groups. Significant differences were set at *p* < 0.05.

## 3. Results

### 3.1. Isolation, Purification and Chemical Components of Polysaccharides

The crude polysaccharide level of *Hericium erinaceus* was obtained by hot water extraction and ethanol precipitation, with the yield reaching 2.735%. The total carbohydrate content of the crude polysaccharides was determined to be 53.36% (including 4.43% reducing sugar). The uronic acid content was 32.56%. After purification by the 5% TCA method, the crude exopolysaccharide (20 mg/mL) of *Hericium erinaceus* was separated and fractionated on DEAE-Cellulose 52 column with gradient elution to give five elution peaks: HEP-1, HEP-2, HEP-3, HEP-4 and HEP-5 (Figure 1A), as detected by the 490 nm absorbance (phenol–sulfuric acid assay). The five fractionated polysaccharides were further purified on the Sephadex G100 column to obtain a single peak (Appendix A). Ultraviolet spectrum of HEP 1–5 was shown in Appendix A. There were no absorption peaks at 280 nm and 260 nm, indicating that the five homogeneous polysaccharides did not contain protein and nucleic acid. 

### 3.2. Structural Characterization of HEPs

The single symmetrical peak presented respectively in Appendix A by HPLC suggested that it was a homogeneous polysaccharide. Based on the calibration curve of dextran standards, the molecular weight of HEP 1–5 is calculated in Table 1.

GC-MS results indicated that HEPs was mainly composed of Fuc, Gal and Glu in a molar ratio of 7.87:68.41:23.72 (Appendix A). Moreover, in this study the precise types of monosaccharide, such as ʟ-fucose, galactopyranose, πᴅ-galactopyranose, πᴅ-galactofuranose and glucopyranose in a molar ratio of 7.87:7.18:15.06:46.17:23.72, were further confirmed. 

IR spectroscopic analysis (Figure 1B) revealed that the broadly stretched intense peak at 3369.64 cm^−1^ was ascribed to the hydroxyl stretching vibration. A distinct C–H group stretching band at 2931.80 cm^−1^ could also be observed. The absorption peaks observed at 1367.53 and 1334.74 cm^−1^ were consistent with the presence of C–H bonds. The absorption peak appeared in 1238.30 cm^−1^ was due to the stretching vibration of S=O. The signals present at 1055.06 and 1045.42 cm^−1^ indicated that HEPs was in the furanose form. Figure 1B shows the IR spectra of HEPs exhibited the absorption at 831.32 and 740.67 cm^−1^, typical for α configuration. 

### 3.3. Methylation Analysis

Methylation analysis by GC-MS of five polysaccharide fractions is shown in Table 2. Based on the result of GC-MS analysis, we deduced that the five fractions had 10 components, including 2,3,4,6-Me4-Glc, 2,3,6-Me3-Glc, 2,3,4-Me3-Glc, 2,4,6-Me3-Glc, 2,4-Me2-Glc, 2,3,4-Me3-Gal, 3,4-Me2-Gal, 2,3,4-Me3-Fuc and 3,4,6-Me3-Fru. These results suggested that the residues of branch structure of HEP-1 were composed of O-3-branched (1→6)-linked-β-ᴅ-Glcp with a few α-ᴅ-glucopyranosyl residues, β-ᴅ-galactofuranosyl residues and α-ʟ-fucoglycosyl residues, which linked to the oxygen of C-2 position of (1→6)-linked-α-ᴅ-Galp backbone. Similarly, HEP-2 and HEP-3 were both composed of a (1→6)-linked-α-ᴅ-Galp backbone with a (1→6)-linked-β-ᴅ-Glcp branch. However, there were some differences in their residues. HEP-2 was composed of →1)-Fruf-(2→1)-α-ᴅ-Glcp, and HEP-3 was composed of →1)-β-ᴅ-Galp -(6→1)-α-ᴅ-Glcp. Moreover, HEP-4 had a backbone of (1→6) linked galactofuranosyl residues substituted at C-3 with a (1→6) linked α-ᴅ-glucopyranose side-chain. HEP-5 had a (1→4)-linked-α-Glc with a α-ᴅ-glucopyranosyl residue. 

### 3.4. NMR Analysis

The ^1^H NMR and ^13^C NMR spectra of HEP-1 are shown in Figure 1C,D. The signals of HEP-1 were mainly distributed in δH 3.0–5.5 (Figure 1C) and δC 60–110 (Figure 1D), which were assigned to the typical distribution of NMR signals of the polysaccharides.

In the ^1^H NMR spectra of HEP-1, the chemical shifts of anomeric H-1 at δ 5.34, 5.33, 5.04, 5.01, 4.46, 4.14, 4.01 and 3.95 ppm were higher or lower than 5.0 ppm, indicating the existence of both α- and β-configurations. From the areas of the anomeric proton signals, it could be speculated that the main peak of ᴅ-glucopyranosyl might be overlapped by D_2_O, where it should be at δ4.78. Moreover, the anomeric proton signals δH5.15, 5.04, 5.01, 4.93, 4.78, 4.76 ppm are assigned to the proton H-1 of α-ʟ-Fucp-(1→, →6) or →2,6)-α-ᴅ-Galp-(1→, →6)-α-ᴅ-Glcp-(1→, →6)-β-ᴅ-Galp(1→, →3)-β-ᴅ-Glcp-(1→, →3,6)-β-ᴅ-Glcp-(1→, respectively. In addition, signals from δH3.3 to δH4.2 ppm are assigned to the H-2 to H-6 protons, which are listed in Appendix A.

In ^13^C-NMR spectrum (Figure 1D), the signals from the anomeric carbons were distributed between 97.88 and 102.99 ppm, which suggested that there were both α (δ95–101 ppm) and β (δ101–105 ppm) anomeric configurations existing in HEP-1. This result was in agreement with that of the 1H NMR spectrum. As shown in Figure 1D, the signals at δC103.96, 103.31,102.99, 100.90, 100.57, 98.21 ppm corresponded to C-1 of α-ʟ-Fucp-(1→, →3) or →3,6)-β-ᴅ-Glcp-(1→, →6)-β-ᴅ-Galp(1→,→2,6)-α-ᴅ-Galp-(1→, →6)-α-ᴅ-Galp-(1→, →6)-α-ᴅ-Glcp-(1→, and multitudes of signals at δC60–90 were attributed to atoms C2-C6 (Appendix A), respectively. By means of 13C and 1H NMR spectra, the entire assignment of the ^13^C and ^1^H chemical shifts of HEP-1 were achieved and is presented in Figure 1E. Correspondingly, the structural features of the other four fractions of HEPs were also analyzed using the NMR spectroscopy. The major ^13^C and ^1^H NMR chemical shifts assigned for HEP-2, HEP-3, HEP-4 and HEP-5 are listed in Appendix A, and the predicted primary structures of them are established in Figure 1F–I. The results also indicated that the backbone was composed of (1→6)-linked α-ᴅ-galactosyl residues, which branched at O-2, or (1→3)-linked glucosyl residues, which branched at O-6.

### 3.5. Effect of HEPs Fractions on Cytotoxicity, Nitric Oxide and Phagocytic Activity in RAW264.7 Cells

To investigate whether the five HEPs fractions have cytotoxic effects on RAW 264.7 cells, a CCK-8 assay was used to ass the effects of the five HEPs fractions in the different concentration on RAW 264.7 cells proliferation activity for 24 h. The results showed that HEP-2 and HEP-3 in the tested concentration (25–800 μg/mL) had no significant effect on the proliferation of RAW 264.7 cells (Figure 2A), indicating that HEP-2 and HEP-3 in the concentration of 25–800 μg/mL had no cytotoxicity on RAW 264.7 cells. HEP-1 and HEP-4 at the concentration of 100, 200 and 400 μg/mL, and HEP-5 at the concentration of 200 μg/mL significantly promoted the proliferation of RAW 264.7 cells. 

Nitric oxide acts as a major mediator of macrophage, preventing the invasion of bacteria and tumor cells. Therefore, we investigated whether HEPs fractions were able to enhance the release of NO (Figure 2B). With the fermentation, the release of NO gradually ascended and then descended under the administration of the five HEPs fractions. Moreover, HEP-1, HEP-2 and HEP-3 significantly secreted more NO than the other groups in RAW264.7 cells.

Phagocytic activity of macrophages, as another major indicator, reflects the functional activation of macrophages. HEP-1 at the concentrations of 50, 100 and 200 μg/mL and HEP-2 at the concentrations of 100, 200 and 400 μg/mL significantly increased the phagocytic activity of macrophages (Figure 2C). These results of the cytotoxicity experiment, the NO production experiment and the phagocytic activity experiment suggested that HEP-1 and HEP-2 were not toxic to RAW264.7 cells and had potential immunomodulatory effects.

### 3.6. Effect of HEP-1 on the Production of Cytokines in RAW264.7/Caco-2 Cells Model

Intestinal mucosa is the first defense barrier of the body and the largest immune organ in the body area. It consists of intestinal epithelial cells and immune cells, which are exposed to a large number of antigens. Caco-2 cells (human colon cancer cells) are similar to differentiated small intestinal epithelial cells in structure and function. We combined Caco-2 cells and RAW264.7 cells to construct an intestinal simulation system to build a co-culture system in vitro (Figure 2D).

To investigate whether HEP-1 triggers immune responses in macrophages, ELISA assays were used to estimate the secretion levels of cytokines in the lower compartment. The results indicated that HEP-1 in the concentration of 100, 200 and 400 μg/mL evidently triggered the production of IL-6, IL-10, IFN-γ and TNF-α compared with the CK group. In addition, HEP-1 in the concentration of 200 μg/mL had the most significant effect on stimulating the secretion levels of IL-6, IL-10, IFN-γ and TNF-α (Figure 3A–D).

### 3.7. Effect of HEP-1 on the Expression of NF-κB, MAPK and PI3K/Akt Pathways

NF-κB, as the pivot of the multiple signaling pathways, plays an important role in the human immune response and cell proliferation. Due to its inactivation, NF-κB distributed in the cytoplasm forms non-covalent binding complexes with p50-p65-IκB trimers. When activated, IκB-α is phosphorylated and dissociated from NF-κB, which is transported to the nucleus and bind to genes with an NF-κB binding site to initiate transcription. NF-κB also activates the expression of IκB gene. The newly synthesized IκB reactivates NF-κB, creating a spontaneous negative feedback loop to reach a balance [26].

To estimate whether the HEP-1 activates the NF-κB signaling pathway, Western blotting was used to detect the nucleus displacement of NF-κB p65 (Figure 4A). The quantified expression results of IκB-α and p65 was calculated by Quantity One (Figure 4D,E). With the concentration of HEP-1, the content of IκB-α and p65 in cytoplasm gradually decreased, and the content of p65 in the nucleus increased compared with the CK group. It showed that HEP-1 in a concentration-dependent manner induced the degradation of IκB-α and nuclear translocation of NF-κB p65 in RAW264.7 cells, and improved the transcriptional activity of NF-κB. The results suggested that the immunoregulatory activities of HEP-1 were probably associated with the NF-κB signaling pathway.

MAPK (mitogen-activated protein kinase) and PI3K/Akt pathways are important transmitter of signals from the cell surface to the inside of the nucleus. Under the stimulus of polysaccharides, they can modulate the release of NO and TNF-α. We examined the expression of p38, JNK, ERK proteins in MAPK pathway and Akt protein in the Akt pathway. The results showed that HEP-1 significantly upregulated the phosphorylation levels of p38, JNK and ERK proteins in the MAPK pathway in a concentration-dependent manner (Figure 4B,F–H). Similarly, the of Akt protein in the PI3K/Akt pathway significantly upregulated (Figure 4C,I). HEP-1 evidently upregulated the phosphorylation level of Akt protein in a concentration-dependent manner. 

### 3.8. The Role of NF-KB, MAPK and PI3K/Akt Pathways in HEP-1 Activation of Macrophages

To further ensure whether HEP-1 stimulates RAW264.7 cells to secrete NO and TNF-α through NF-κB, MAPK, and PI3K/Akt signaling pathways, we used specific inhibitors, such as NF-κB pathway inhibitors (BAY11-7082 for IκB-α), MAPK pathway inhibitors (SB203580 for p38, SP600125 for JNK, PD98059 for ERK) and Akt pathway inhibitors (LY294002 for Akt) to block the signaling pathways, respectively. RAW264.7 cells were pretreated by inhibitors for 1 h and stimulated by HEP-1 for 24 h. The content of NO and TNF-α in the lower compartment of RAW264.7/Caco-2 cells model was determined. The secretion levels of NO and TNF-α are shown in the histogram (Figure 3E,F). The results revealed that the NF-κB pathway inhibitor BAY11-7082 almost completely inhibited HEP-1-induced secretion of NO and TNF-α. The inhibitors including SP600125, PD98059 and LY294002 inhibited supernatant-induced secretion of NO and TNF-α to varying degrees, while p-p38 inhibitor SB203580 had no effect on HEP-1-induced secretion of NO. These results indicated that NF-κB was an important transcription factor modulating HEP-1 to induce the release of NO and TNF-α in RAW264.7 cells.

We further examined the protein expression of IκB-α and p65 in the NF-κB pathway, of p38, JNK and ERK in the MAPK pathway and of Akt in the PI3K/Akt pathway using Western blotting (Figure 5A–E). The quantified expression of those proteins was calculated by Quantity One (Figure 5F,J). The results showed that five pathway inhibitors almost completely inhibited the HEP-1-induced phosphorylation of related proteins. It suggested that the three pathways are involved in HEP-1-induced immune responses. Based on the results, we drew a figure on immunomodulatory mechanism for HEP-1-induced macrophage activation (Figure 6).

## 4. Discussion

The polysaccharides from the *Hericium erinaceus* mushroom were first reported by Mccracken and Dodd in 1971 [27]. After that, a large number of studies have been carried out, aimed at studying their isolation method, structure features as well as biological activities. Two polysaccharides (AF2S-2, BF2S-2) were isolated from fruiting bodies of *H. erinaceus*, which consisted of a backbone of β-(l→6)-linked ᴅ-glucopyranosyl residues with β-(1→3) and β-(l→6) glucosidic linkages [28]. There were some differences in the chemical composition between the fruiting bodies and mycelia. While both hfp-1 from fruiting bodies and hmp-2 from mycelia were homogeneous, hfp-1 was composed of arabinose, mannose, galactose and glucose in a molar ratio of 0.12:0.04:1.00:0.71 and hmp-2 was composed of arabinose, xylose, mannose, galactose and glucose in a molar ratio of 0.25:0.41:0.31:1.00:0.29. Hfp-1 had β-glycoside linkage, but hmp-2 did not [12]. In our study, we isolated five polysaccharides from the fruiting bodies of *H. erinaceus*, which had the similar backbone and glucosidic linkages with the above report, but the composition and ratio of monosaccharides were different.

*Hericium erinaceus* polysaccharides, belonging to mushroom polysaccharides, are a great immunomodulator [29]. However, most of the studies on *Hericium erinaceus* polysaccharides have focused on the protective effect on gastric mucosal injury. There is limited information on their immunomodulatory activity. Over the past several years, studies on the structure–activity of mushrooms polysaccharides have shown the relatedness between structural diversity and immunestimulatory activity. Mannogalacta purified from the fruiting bodies of hybrid mushrooms could activate macrophage by NO production and stimulate in vitro splenocyte and thymocyte [30]. Heteropolysaccharide isolated from *Boletus edulis* could significantly increase the spleen and thymus indices, stimulate splenocytes proliferation and augment natural killer cell and cytotoxic T-lymphocyte activities in the spleen and promote the secretion of the cytokines IL-2 and TNF-α in vivo [31]. In this study, HEP-1, isolated from *Hericium erinaceus* polysaccharides, is a kind of galactoglucan. HEP-1 exhibited immunestimulatory activity by stimulating the production of NO and the secretion of TNF-α and cytokines in vitro. Similarly, galactoglucan from the fruiting bodies of *Lepista sordida* could activate murine macrophages and elevated NO production and TNF-α secretion in vitro [32]. These polysaccharides are of different origin but have similar structures and immunomodulatory activities.

Although polysaccharides had different structures, they both exhibited immunomodulatory activity. It seems to indicate that the immune response is non-specific or partially non-specific. Some other characteristics of polysaccharides may play an important role in stimulating immune responses, such as molecular weight, solubility, branching configuration rather than chemical structures. High molecular weight (500–2000 kDa) mushroom polysaccharides appear to be more immunostimulatory activity compared to low molecular weight polysaccharides [33,34]. For example, the high molecular weight fraction from *Grifola frondosa* (<800 kDa) had the greater immunomodulatory activity than the lower molecular weight fraction [35]. The monosaccharide composition of polysaccharides had the correlation with immunostimulatory activities. Polysaccharides composed of mannose, rhamnose and fucose are known to show better immunomodulatory potency [36]. For instance, polysaccharides containing Mannose have protective effects against some virulent pathogens [37]. In addition, solubility, branching configuration, helical conformation and chemical modification also can determine the immunomodulatory functions [38]. Polysaccharides with water-soluble, debranch, random coil conformation and appropriate chemical modification are also capable of showing significant immunomodulatory activity. In this study, HEP-1 is a water-soluble polysaccharide, and have high molecular weight (2120 kDa). Monosaccharides composition includes fucose which has better immunomodulatory potency. These characteristics play an essential role of immunostimulatory activity of HEP-1.

When infection occurs, the innate immune system is the first line of defense for the system, providing a barrier for pathogens or other foreign particles to help slow the infection, and then forming an acquired immune response. Phagocytes in the innate immune system are activated by a pathogen that enters the host cell and then conduct the process of phagocytosis [39]. Macrophages are particularly responsible for the production of TNF-α, IL-1, IL-6, IL-10 and IL-12, which ultimately leads to immune stimulation. Cytokines TNF-α and IL-6 are major immune mediators. TNF-α plays an important role in the immune stimulation. It can inhibit tumor, promote factors and also induce the release of some other cytokines [40]. IL-6 plays an important role in delivering signals from innate immune response to adaptive immune response [41]. In our study, HEP-1 treatment promoted the production of NO and expression of pro-inflammatory cytokines such as TNF-α, IL-6 and IFN-γ in RAW 264.7 cells. The release of those immune cytokines is involved with the activation of multiple signaling pathways, such as the NF-κB pathway, MAPK pathway and PI3K/Akt pathway. When these signal pathways are activated, substrates and transcription factors are phosphorylated, and immune cytokines are secreted. In turn the production of these cytokines induces the expression of genes responsible for the immune cytokines. Our findings revealed that HEP-1 treatment leads to the phosphorylation of IκB-α, p38, JNK, ERK and Akt in RAW 264.7 cells, indicating that HEP-1 activated the NF-κB pathway, MAPK pathway and PI3K/Akt pathway which leads to the phosphorylation of IκB-α, p38, JNK, ERK and Akt resulting in the production of TNF-α, IL-6 and IFN-γ. HEP-1 displayed remarkable immunomodulatory activity. 

## 5. Conclusions

Our findings provide a better understanding of the structural characterization and the molecular mechanism of HEP-mediated murine macrophage activation and portray the possibility of HEPs as edible and pharmaceutical agents with multidimensional applications. 

## Figures and Tables

**Figure 1 nutrients-14-03721-f001:**
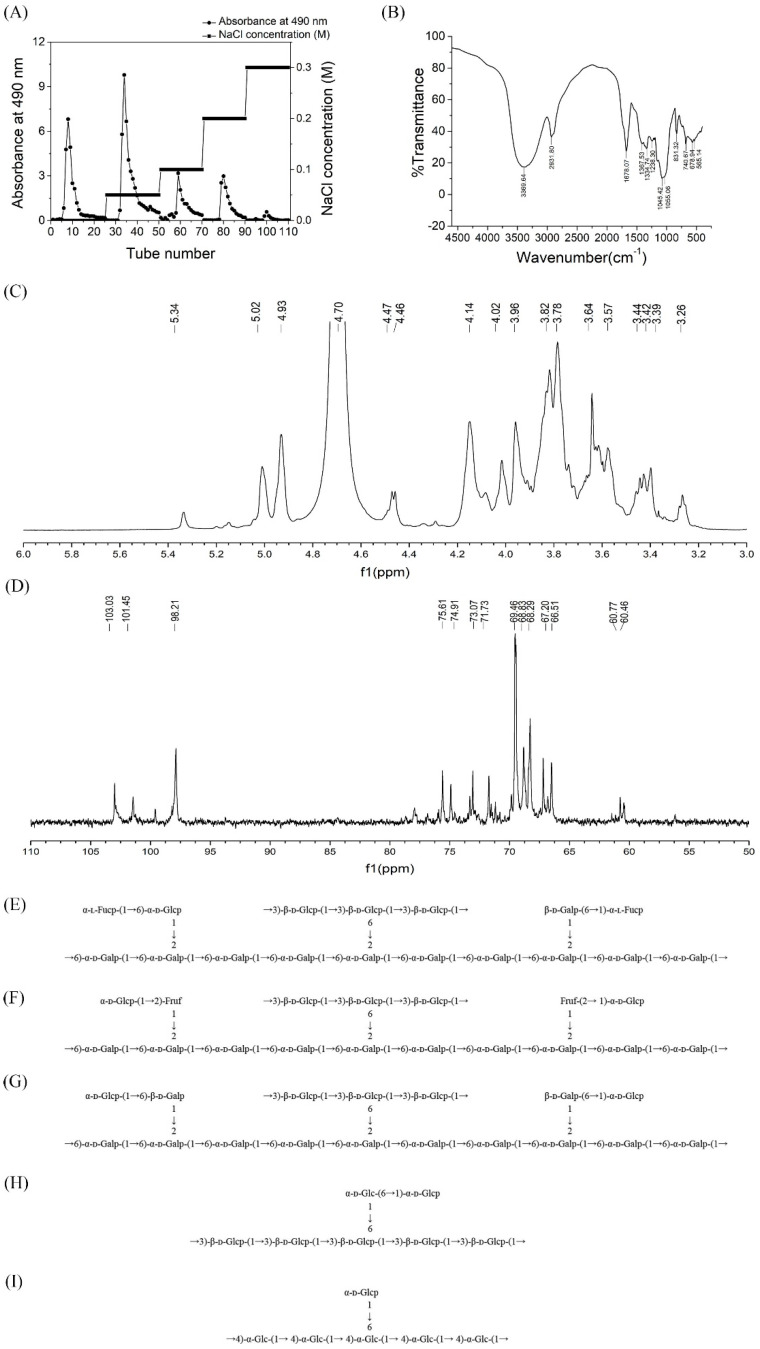
Structural analysis of polysaccharides from *Hericium erinaceus* fruiting bodies. (**A**) Gradient elution curve of crude HEP on DEAE Cellulose-52 chromatography column. (**B**) FT-IR spectrum of HEP in the range of 4000–500 cm^−1^. (**C**) ^1^H-NMR and (**D**) ^13^C-NMR spectrum of HEP-1 isolated from *Hericium erinaceus*. Predicted structure of (**E**) HEP-1, (**F**) HEP-2, (**G**) HEP-3, (**H**) HEP-4 and (**I**) HEP-5 produced by the fungus *Hericium erinaceus*.

**Figure 2 nutrients-14-03721-f002:**
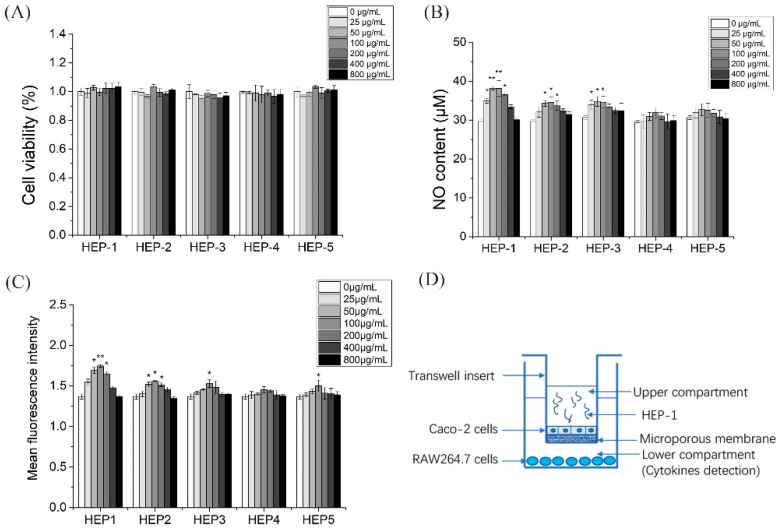
Effects of HEP-1, HEP-2, HEP-3, HEP-4 and HEP-5 in RAW264.7 cells. (**A**) Cell viability. (**B**) NO production. (**C**) Phagocytic activity. (**D**) The co-culture model of RAW264.7/Caco-2 cells. Data are presented as mean ± SD (*n* = 3). * *p* < 0.05, ** *p* < 0.01 compared with the CK group.

**Figure 3 nutrients-14-03721-f003:**
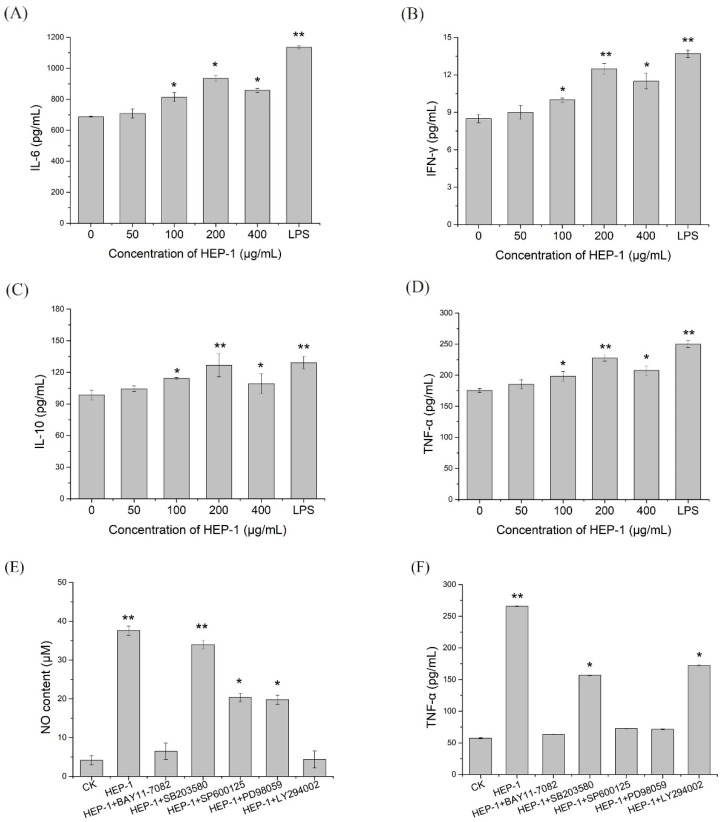
Effects of HEP-1 in RAW264.7/Caco-2 cells model. (**A**) IL-6. (**B**) IFN-γ. (**C**) IL-10. (**D**) TNF-α. After the treatment of inhibitors including BAY11-7082, SB203580, SP600126, PD98095 and LY294002, the release of (**E**) NO and (**F**) TNF-α in NF-κB, MAPK and PI3K/Akt signaling pathways were detected. Data are presented as mean ± SD (*n* = 3). * *p* < 0.05, ** *p* < 0.01 compared with the CK group.

**Figure 4 nutrients-14-03721-f004:**
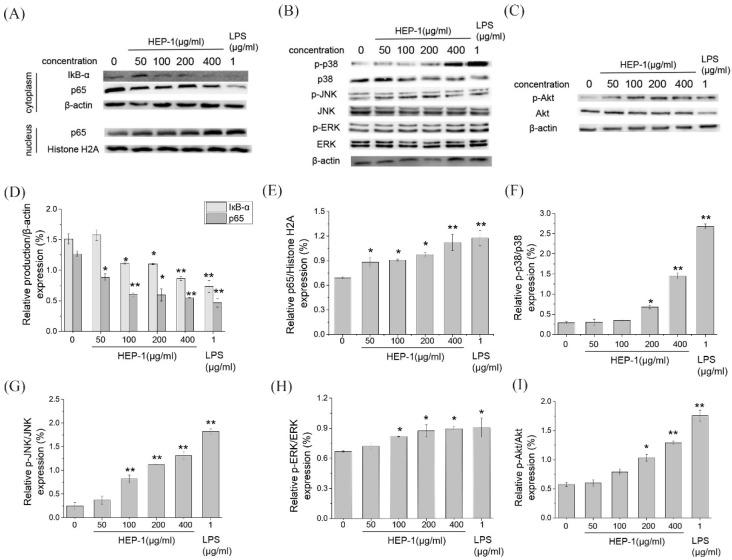
Immunomodulatory activities of HEP-1 detected by RAW264.7/Caco-2 cells model. Effect of HEP-1 on cytoplasm/nuclear (**A**) NF-κB (p65), (**B**) MAPK, (**C**) PI3K/Akt signaling pathways. The quantified expression of (**D**) IκB-α or p65/β-action on cytoplasm and (**E**) p65/Histone H2A on nucleus. The quantified expression of (**F**) p-P38/P38, (**G**) p-JNK/JNK, (**H**) p-ERK/ERK. The quantified expression of p-Akt/Akt (**I**). Cells were incubated with HEP-1 (0, 50, 100, 200, 400 μg/mL) or LPS (1 μg/mL) for 3 h. Data are presented as mean ± SD (*n* = 3). * *p* < 0.05, ** *p*<0.01 compared with the CK group.

**Figure 5 nutrients-14-03721-f005:**
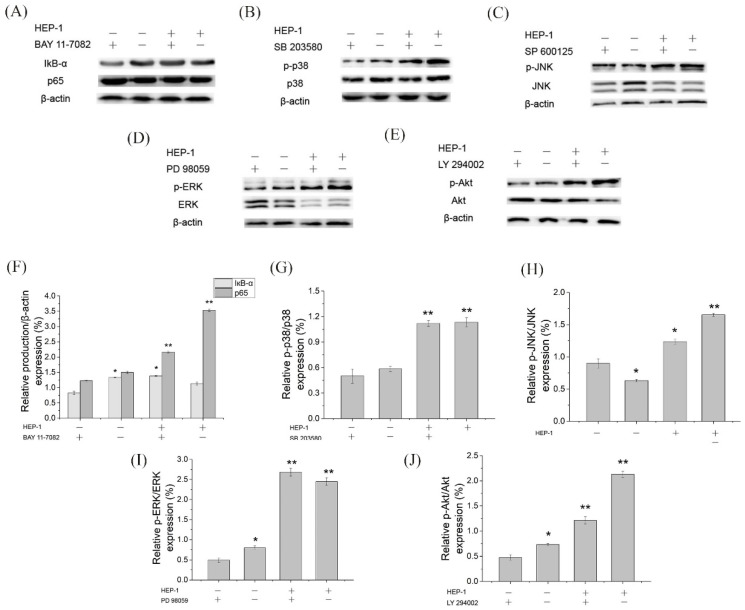
Effects of inhibitors of NF-κB, MAPK and PI3K/Akt signaling pathways on the HEP-1-induced phosphorylation of related proteins in RAW264.7/Caco-2 cells model. Inhibitory effects of (**A**) BAY11-7082, (**B**) SB203580, (**C**) SP600126, (**D**) PD98095 and (**E**) LY294002 on the NF-κB, MAPK and PI3K/Akt signaling pathways, and the quantified expression of (**F**) IκB-α or p65/β-action, (**G**) p-P38/P38, (**H**) p-JNK/JNK, (**I**) p-ERK/ERK and (**J**) p-Akt/Akt. Data are presented as mean ± SD (*n* = 3). * *p* < 0.05, ** *p* < 0.01 compared with the CK group.

**Figure 6 nutrients-14-03721-f006:**
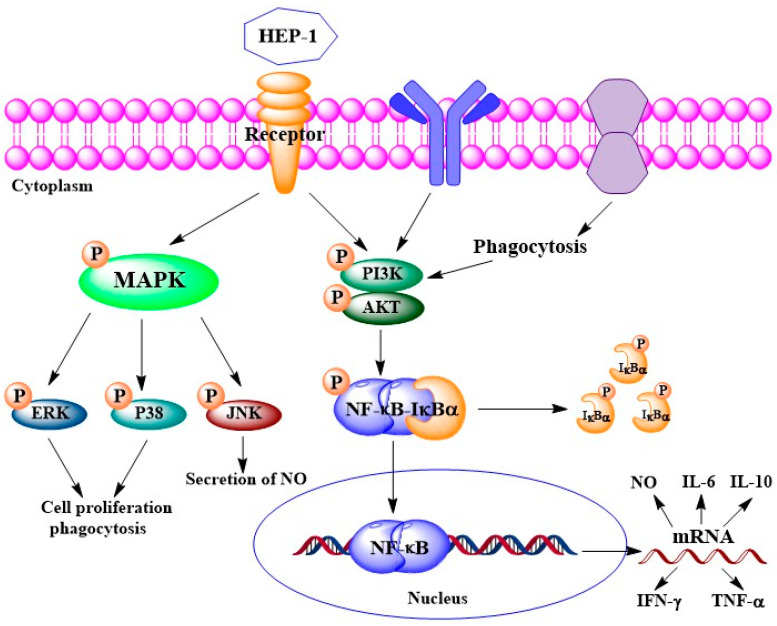
Potential immunomodulatory mechanism for HEP-1-induced macrophage activation.

**Table 1 nutrients-14-03721-t001:** Molecular weight of the HEP 1–5.

Sample	Average Retention Time	Average Molecular Weight (Da)
HEP-1	22.05	2.12 × 10^6^
HEP-2	23.26	9.27 × 10^5^
HEP-3	26.37	1.10 × 10^5^
HEP-4	28.87	1.99 × 10^4^
HEP-5	29.75	1.09 × 10^4^

**Table 2 nutrients-14-03721-t002:** GC-MS results of methylation analyses of polysaccharides HEP-1, HEP-2, HEP-3, HEP-4 and HEP-5.

Methylation Sugar	Linkages	Molar Ratios				
		HEP-1	HEP-2	HEP-3	HEP-4	HEP-5
2,3,4,6-Me_4_-Glc	T→	-	1.09	0.23	0.06	0.04
2,3,6-Me_3_-Glc	1→4	-	-	-	-	0.24
2,3-Me_2_-Glc	1→4,6	-	-	-	-	0.09
2,3,4-Me_3_-Glc	1→6	0.85	-	-	0.04	-
2,4,6-Me_3_-Glc	1→3	2.07	2.99	0.96	0.37	-
2,4-Me_2_-Glc	1→3,6	0.28	0.21	0.11	0.10	-
2,3,4-Me_3_-Gal	1→6	6.18	5.37	1.99	-	-
3,4-Me_2_-Gal	1→2,6	0.73	0.61	0.21	-	-
2,3,4-Me_3_-Fuc	T→	1.33	-	-	-	-
3,4,6-Me_3_-Fru	1→2	-	0.98	-	-	-

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
