# Peer review of "Polysaccharides from Hericium erinaceus Fruiting Bodies: Structural Characterization, Immunomodulatory Activity and Mechanism"

_nutrients, 2022, doi:10.3390/nu14183721_

Round 1
Reviewer 1 Report
General comments: This study approaches the characterization of different carbohydrate fraction from H. erinaceus, which constitute a strength. It could be convenient to define potential physiological concentrations achievable after mushroom consumption or compare those with similar carbohydrates used in by the farma industry. My major concern refers to the term 'immunomodulatory'. This activity implies much more than cytokine regulation. My recommendation is not to use this term throughout. The discussion section could be completed with brief comments concerning the potential context where HEP-1 could be used, because activated macrophages can either improve or worsen diseases.
Line 31: how these activities have been demonstrated? In vivo, preclinical, cell culture or laboratory?. Please, indicate this general classification to help the reader understand the potential of H. erinaceus. It could also be convenient to state the novelty of the present manuscript in relation to previous studies.
Line 47: Structural differences do not seem so determinant to focus on immunomodulatory features. For example, innate immune ‘Toll-like’ receptors recognize several different structures to signal via the same receptor. Moreover, nutrient availability clearly will affect immune cell function. In my opinion, authors should justify better the study of potential activity for the carbohydrate fractions.
Line 61: TLR4 signaling is closely associated to NFkB pathway. But NFkB is activated by many other pathways. Specific TLR4 inhibitors have not been used in the study. Please, rewrite these sentences throughout.
Line 163: Were Caco-2 cells used under confluency and full metabolic differentiation or without cell monolayer development for the studies?. The significance of the study results completely different.
Figure 1. Panel B-D: it could be interesting to show the major molecular groups attributable to the wave detected.
Figure 2. Panel C: How do authors measure neutral red fluorescence?. Not absorbance?; Panel D, activities appear relatively similar for all carbohydrate fractions. Is it possible to introduce additional comments based on the different composition of the fraction?. Also, what is the carbohydrate concentration found in the basolateral media after exposure to Caco-2 cells?.
Line 320: ‘…Nitric oxide as a major mediator of macrophage…’…what?. It seems that some words defining action, effects,etc.. are missing.
Line 324: ‘…accelerated…’. Authors do not provide the time-course for NO production. Please, rewrite this word.
Line 336-338: ‘…Caco-2 cells (human colon cancer cells) are similar to differentiated small intestinal epithelial cells in…’. This sentence is true when used after 13-15 days growing to reach confluence and cell polarization. However, the conditions used in this study do not seem to allow complete cell differentiation and polarization. Please, rewrite or delete this sentence.
Line 347: The cytokines assayed play different roles, worsening or improving, metabolic and/or immune diseases. For example, IL-6 is widely believed to play a role in the progression and severity of many forms of cancer (Sci Rep 5, 11394, 2015). However, IL-6 exerts beneficial metabolic effects in obesity (2017, Cell Reports 19, 267–280). Macrophages play determinant roles in both diseases. Authors should refer to this type of effects in order to propose any potential immunomodulatory effects, and the context.
Line 380: what is the rationale to use different inhibitors?. Please, state the reason or different molecular target.
Figure 6. Authors assume that TLR4 is the main receptor for HEP-1. However, the active site of this receptor is an hydrophobic site and changes in the expression or intracellular relocation of the receptor has not been monitored.
Line 443: ‘…potent immune-modulator…’. Based on…?. Please, state references.
Author Response
General comments: This study approaches the characterization of different carbohydrate fraction from H. erinaceus, which constitute a strength. It could be convenient to define potential physiological concentrations achievable after mushroom consumption or compare those with similar carbohydrates used in by the farma industry. My major concern refers to the term 'immunomodulatory'. This activity implies much more than cytokine regulation. My recommendation is not to use this term throughout. The discussion section could be completed with brief comments concerning the potential context where HEP-1 could be used, because activated macrophages can either improve or worsen diseases.
Line 31: how these activities have been demonstrated? In vivo, preclinical, cell culture or laboratory?. Please, indicate this general classification to help the reader understand the potential of H. erinaceus. It could also be convenient to state the novelty of the present manuscript in relation to previous studies.
Response: Thanks for pointing this out. As suggested, general classification have been added (Line 50 -55).
Line 47: Structural differences do not seem so determinant to focus on immunomodulatory features. For example, innate immune ‘Toll-like’ receptors recognize several different structures to signal via the same receptor. Moreover, nutrient availability clearly will affect immune cell function. In my opinion, authors should justify better the study of potential activity for the carbohydrate fractions.
Response: Thanks for pointing this out. Indeed, structure is not a decisive influence on immunomodulatory activity. Many other characteristics are also important. We've added a discussion about it in Discussion (Line 538-558). And about the nutrients, immune cells were cultured in an DMEM medium supplemented with 10% (v/v) fetal bovine serum. The immune cells can get plenty of nutrients. As you mentioned, the study mainly investigated the immunomodulatory activity of polysaccharide fractions HEP-1 and mechanisms.
Line 61: TLR4 signaling is closely associated to NFkB pathway. But NFkB is activated by many other pathways. Specific TLR4 inhibitors have not been used in the study. Please, rewrite these sentences throughout.
Response: Thanks for pointing this out. The TLR4 signaling mentioned here is the results of other researchers' studies, not this study.
Line 163: Were Caco-2 cells used under confluency and full metabolic differentiation or without cell monolayer development for the studies?. The significance of the study results completely different.
Response: Thanks for pointing this out. The cultivation and treatment methods of Caco-2 were not written in detail. Now they have been completed (Line 202-209). Caco-2 cells were used under they differentiating into dense polar monolayers on the membrane, because polarized Caco-2 cells can express some structural and functional features of intestinal mucosal epithelial cells.
Figure 1. Panel B-D: it could be interesting to show the major molecular groups attributable to the wave detected.
Response: Thanks for pointing this out. Panel B: Characteristic absorption peaks of polysaccharides in the mid-infrared spectrum (4000-650 cm-1 ) were needed to be analyzed, so the full spectrum (4000-650 cm-1 ) was displayed. This ensures that we didn't miss any characteristic absorption peaks. Panel C and D have been enlarged for easy viewing.
Figure 2. Panel C: How do authors measure neutral red fluorescence?. Not absorbance?; Panel D, activities appear relatively similar for all carbohydrate fractions. Is it possible to introduce additional comments based on the different composition of the fraction?. Also, what is the carbohydrate concentration found in the basolateral media after exposure to Caco-2 cells?.
Response: Thanks for pointing this out. Panel C: We measure neutral red fluorescence by the method in 2.4.4. The absorbance was detected by a microplate ELISA reader at 540 nm. The co-culture system in Panel D was designed to simply mimic that carbohydrate fractions stimulate immune cells to secrete immune factors by stimulating intestinal epithelial cells to produce signals, rather than studying the direct effects of carbohydrate fractions on immune cells. The detection of the carbohydrate concentration in the basolateral media may not be consistent with this purpose. According to the result in Panel A B C, HEP-1 were the best. So HEP-1 was selected for further study.
Line 320: ‘…Nitric oxide as a major mediator of macrophage…’…what?. It seems that some words defining action, effects,etc.. are missing.
Response: Thanks for pointing this out. It has been revised. This may be caused by different expression habits.
Original: “Nitric oxide as a major mediator of macrophage prevents the invasion of bacteria and tumor cells.”
Revised (Line 346-347): “Nitric oxide acts as a major mediator for macrophages, preventing the invasion of bacteria and tumor cells.”
Line 324: ‘…accelerated…’. Authors do not provide the time-course for NO production. Please, rewrite this word.
Response: Corrected. Thanks.
Original: “Moreover, HEP-1, HEP-2 and HEP-3 significantly accelerated the secretion of NO in RAW264.7 cells compared to the other groups.”
Revised (Line 350-351): “Moreover, HEP-1, HEP-2 and HEP-3 significantly secreted more NO than the other groups in RAW264.7 cells.”
Line 336-338: ‘…Caco-2 cells (human colon cancer cells) are similar to differentiated small intestinal epithelial cells in…’. This sentence is true when used after 13-15 days growing to reach confluence and cell polarization. However, the conditions used in this study do not seem to allow complete cell differentiation and polarization. Please, rewrite or delete this sentence.
Response: Thanks for pointing this out. Because the cultivation and treatment methods of Caco-2 were not written in detail, there was a misunderstanding of this study. Now they have been completed (Line 202-209). The sentence “Caco-2 cells (human colon cancer cells) are similar to differentiated small intestinal epithelial cells in structure and function.” remains.
Line 347: The cytokines assayed play different roles, worsening or improving, metabolic and/or immune diseases. For example, IL-6 is widely believed to play a role in the progression and severity of many forms of cancer (Sci Rep 5, 11394, 2015). However, IL-6 exerts beneficial metabolic effects in obesity (2017, Cell Reports 19, 267–280). Macrophages play determinant roles in both diseases. Authors should refer to this type of effects in order to propose any potential immunomodulatory effects, and the context.
Response: Thanks for pointing this out. As suggested, some discussion about this are added (Line 561-587).
Line 380: what is the rationale to use different inhibitors?. Please, state the reason or different molecular target.
Response: Thanks for pointing this out. We have stated the targets of the different inhibitors in the text.
Original: “We used specific inhibitors, such as NF-кB pathway inhibitors (BAY11-7082), MAPK pathway inhibitors (SB203580, SP600125, PD98059) and Akt pathway inhibitors (LY294002) to block the signaling pathways, respectively.”
Revised (Line 447-449): “We used specific inhibitors, such as NF-кB pathway inhibitors (BAY11-7082 for IκB-α), MAPK pathway inhibitors (SB203580 for p38, SP600125 for JNK, PD98059 for ERK) and Akt pathway inhibitors (LY294002 for Akt) to block the signaling pathways, respectively. ”
Figure 6. Authors assume that TLR4 is the main receptor for HEP-1. However, the active site of this receptor is an hydrophobic site and changes in the expression or intracellular relocation of the receptor has not been monitored.
Response: Thanks for pointing this out. We change “TLR4” into “Receptor”. Make it more rigorous.
Line 443: ‘…potent immune-modulator…’. Based on…?. Please, state references.
Response: Thanks for pointing this out. The sentence has been modified. References have been added.
Original: ‘…potent immune-modulator…’.
Revised (Line 520-521): ‘…great immune-modulator…’.

Reviewer 2 Report
Recently, the polysaccharide from H. erinaceus (HEP) is supposed to be one of the major bioactive compounds, which garnered widespread attention in seeking innovative applications for precision nutrition. Both in vitro and in vivo studies revealed that HEP has potential antioxidant and anti-aging, gastrointestinal protecting, anti-tumor, hypolipidemic and hypoglycemic, immune-enhancing, neuroprotective, and hepatoprotective activities... The current research prospects of HEP from Yang et al provided a better understanding of the structural characterization and the molecular mechanism of HEP-mediated murine macrophage activation and portray the possibility of HEPs as edible and pharmaceutical agents which may be helpful for the healthy application of HEP further. Overall, this is an interesting manuscript, there are still some limitations that need to be well enhanced.
1. It would be good to provide that immunologbloting before trimming as Supplement. Regarding this, Fig. 4H had no difference among the testing
2. The Refs need to be updated, some recent ones are not included. Ex. 10.1016/j.jfutfo.2022.03.007
3. Discussion chapter need to be re-written to highlight their findings in comparison previous reported in term of getting a better understanding of the current findings and some potential molecular mechanisms as immunomodulatory agent potential in in dietary supplements application.
4. Fig. 1 is really low quality, especially in C, D, please revise it accordingly
5. Many typing errors were found and formal English writing needed to be improved throughout the manuscript. Ex: Line 172: ..cells/mL in a volume of 100 L in 96-well plate, how did you do? Also, typically concentration of 1 × 105 cells/mL in a well in a 96-well plate, seemed to be too dense, and able to make their results not stable.
7. How cell in WB methods need to provide detailed
Author Response
Recently, the polysaccharide from H. erinaceus (HEP) is supposed to be one of the major bioactive compounds, which garnered widespread attention in seeking innovative applications for precision nutrition. Both in vitro and in vivo studies revealed that HEP has potential antioxidant and anti-aging, gastrointestinal protecting, anti-tumor, hypolipidemic and hypoglycemic, immune-enhancing, neuroprotective, and hepatoprotective activities... The current research prospects of HEP from Yang et al provided a better understanding of the structural characterization and the molecular mechanism of HEP-mediated murine macrophage activation and portray the possibility of HEPs as edible and pharmaceutical agents which may be helpful for the healthy application of HEP further. Overall, this is an interesting manuscript, there are still some limitations that need to be well enhanced.
1.It would be good to provide that immunologbloting before trimming as Supplement. Regarding this, Fig. 4H had no difference among the testing
Response: Thanks for pointing this out. There was a significant difference in Fig. 4H. The “ * ” in Fig. 4H presented p<0.05. I'm willing to provide the immunologbloting if the magazine need it.
2.The Refs need to be updated, some recent ones are not included. Ex. 10.1016/j.jfutfo.2022.03.007
Response: Thanks for pointing this out. As suggested, some recent Refs have been added. (Line 521, 527,531,535, 570 )
3.Discussion chapter need to be re-written to highlight their findings in comparison previous reported in term of getting a better understanding of the current findings and some potential molecular mechanisms as immunomodulatory agent potential in in dietary supplements application.
Response: Thanks for pointing this out. As suggested, Discussion chapter have been re-written. (Line 506-587)
4.Fig. 1is really low quality, especially in C, D, please revise it accordingly
Response: Thanks for pointing this out. As suggested, We've enlarged Fig. 1 C and D.
5.Many typing errors were found and formal English writing needed to be improved throughout the manuscript. Ex: Line 172: ..cells/mL in a volume of 100 L in 96-well plate, how did you do? Also, typically concentration of 1 × 105 cells/mL in a well in a 96-well plate, seemed to be too dense, and able to make their results not stable.
Response: Corrected. Thanks. We've changed “L” to “μL”, and “ 96-well” to “6-well” .
Original: “..cells/mL in a volume of 100 L in 96-well plate,”
Revised (Line 177-178): “..cells/mL in a volume of 100 μL in 6-well plate,”
6.How cell in WB methods need to provide detailed
Response: Thanks for pointing this out. How cell in WB methods has been provided in 2.4.6., including the methods of treating cells with or without specific inhibitors (Line 219-232).
